# Trial matching: capturing variability with data-constrained spiking neural networks

**Christos Sourmpis,  Carl C.H. Petersen,  Wulfram Gerstner,  Guillaume Bellec**
Brain Mind Institute,
School of Computer and Communication Sciences and School of Life Sciences,
École Polytechnique Fédérale de Lausanne (EPFL),
CH-1015 Lausanne, Switzerland,
{firstname.lastname}@epfl.ch

## Abstract

Simultaneous behavioral and electrophysiological recordings call for new methods to reveal the interactions between neural activity and behavior. A milestone would be an interpretable model of the co-variability of spiking activity and behavior across trials. Here, we model a mouse cortical sensory-motor pathway in a tactile detection task reported by licking with a large recurrent spiking neural network (RSNN), fitted to the recordings via gradient-based optimization. We focus specifically on the difficulty to match the trial-to-trial variability in the data. Our solution relies on optimal transport to define a distance between the distributions of generated and recorded trials. The technique is applied to artificial data and neural recordings covering six cortical areas. We find that the resulting RSNN can generate realistic cortical activity and predict jaw movements across the main modes of trial-to-trial variability. Our analysis also identifies an unexpected mode of variability in the data corresponding to task-irrelevant movements of the mouse.

## 1  Introduction

Over the past decades, there has been a remarkable advancement in neural recording technology. Today, we can simultaneously record hundreds, even thousands, of neurons with millisecond time precision. Coupled with behavior measurements, modern experiments enable us to better understand how brain activity and behavior are intertwined [1]. In these experiments, it is often observed that even well-trained animals respond to the same stimuli with considerable variability. For example, mice trained on a simple tactile detection task occasionally miss the water reward [2], possibly because of satiation, lack of attention or neural noise. It is also clear that there is additional uncontrolled variability in the recorded neural activity [3, 4, 5] induced for instance by a wide range of task-irrelevant movements.

Our goal is to reconstruct a simulation of the sensory-motor circuitry driving the variability of neural activity and behavior. To understand the generated activity at a circuit level, we develop a generative model which is biologically interpretable: all the spikes are generated by a recurrent spiking neural network (RSNN) with hard-biological constraints (i.e. the voltage and spiking dynamics are simulated with millisecond precision, neurons are either inhibitory or excitatory, spike transmission delay takes $2 - 4$ ms)[1].

First contribution, we make a significant advance in the simulation methods for data-constrained RSNNs. While most prior works [6, 7, 8] were limited to single recording sessions, our model is

---

[1]Our code is available at github.com/EPFL-LCN/pub-sourmpis2023-neurips (code DOI hosted by zenodo).

constrained to spike recordings from 28 sessions covering six cortical areas. The resulting spike-based model enables a data-constrained simulation of a cortical sensory-motor pathway (from somatosensory to motor cortices responsible for the whisker, jaw and tongue movements). As far as we know, our model is the first RSNN model constrained to multi-session recordings with automatic differentiation methods for spiking neural networks [8, 9, 10].

Second contribution, using this model we aim to pinpoint the circuitry that induces variability in behavior (asking for instance what circuit triggers a loss of attention). Towards this goal, we identify an unsolved problem: "how do we enforce the generation of a realistic distribution of neural activity and behavior?" To do this, the model is fitted jointly to the recordings of spiking activity and movements to generate a realistic trial-to-trial co-variability between them. Our technical innovation is to define a supervised learning loss function to match the recorded and generated variability. Concretely the *trial matching* loss function is the distance between modeled and recorded distributions of neural activity and movements. It relies on recent advances in the field of optimal transport [11, 12, 13] providing notions of distances between distributions. In our data-constrained RSNN, *trial matching* enables the recovery of the main modes of trial-to-trial variability which includes the neural activity related to instructed behavior (e.g. miss versus hit trials) and uninstructed behavior like spontaneous movements.

**Related work** While there is a long tradition of data fitting using the leaky integrate and fire (LIF) model, spike response models [14] or generalized linear models (GLM) [6], most of these models were used to simulate single neuron dynamics [15, 16] or small networks with dozens of neurons recorded in the retina and other brain areas [6, 7, 8, 17]. A major drawback of those fitting algorithms was the limitation to a single recording session. Beyond this, researchers have shown that FORCE methods [18] could be used to fit up to 13 sessions with a large RSNN [17, 19, 20]. But in contrast with back-propagation through time (BPTT) in RSNNs [9, 10, 21], FORCE is tied to the theory of recursive least squares making it harder to combine with deep learning technology or arbitrary loss functions. We know only one other study where BPTT is used to constrain RSNN to spike-train recordings [8] but this study was limited to a single recording session.

Regarding generative models capturing trial-to-trial variability in neural data, many methods rely on trial-specific latent variables [22, 23, 24, 25, 26]. This is often formalized by abstracting away the physical interpretation of these latent variables using deep neural networks (e.g. see LFADS [22] or spike-GAN [27]) but our goal is here to model the interpretable mechanisms that can generate the recorded data. There are hypothetical implementations of latent variables in RSNNs, most commonly, latent variables can be represented as the activity of mesoscopic populations of neurons [25], or linear combinations of the neuronal activity [26, 28, 29]. These two models assume respectively an implicit grouping of the neurons [25] or a low-rank connectivity matrix [26, 28, 29]. Here, we want to avoid making any structural hypothesis of this type a priori. We assume instead that the variability is sourced by unstructured noise (Gaussian current or Poisson inputs) and optimize the network parameters to transform it into a structured trial-to-trial variability (e.g. multi-modal distribution of hit versus miss trials). The optimization therefore decides what is the network mechanism that best explains the trial-to-trial variability observed in the data.

This hypothesis-free approach is made possible by the *trial matching* method presented here. This method is complementary to previous optimization methods for generative models in neuroscience. Many studies targeted solely trial-averaged statistics and ignored single-trial activity, for instance methods using the FORCE algorithm [17, 19, 20, 30, 31], RSNN methods using back-propagation through time [8] and multiple techniques using (non-interpretable) deep generative models [32]. There exist other objective functions which can constrain the trial-to-trial variability in the data, namely: the maximum likelihood principle [6, 15] or spike-GANs [27, 33]. We illustrate however in the discussion section why these two alternatives are not a straightforward replacement for the *trial matching* loss function with our interpretable RSNN generator.

## 2   Large data-constrained Recurrent Spiking Neural Network (RSNN)

This paper aims to model the large-scale electrophysiology recordings from [2], where they recorded 6,182 units from 12 areas across 18 mice [2]. All animals in this dataset were trained to perform the whisker tactile detection task described in Figure 1: in 50% of the trials (the GO trials), a whisker is

---

[2]of which we use 3,810 units from 6 cortical areas in this study

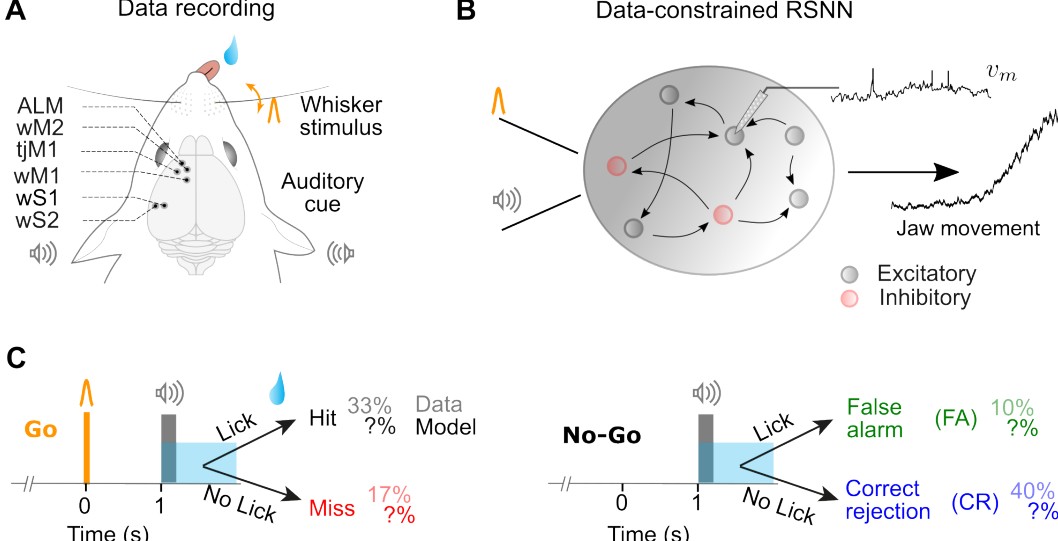

Figure 1: **Modeling trial-variability in electrophysiological recordings.** **A**. During a delayed whisker detection task, the mouse should report the sensation of a whisker stimulation by licking to obtain a water reward. Neural activity and behavior of the mouse are recorded simultaneously. **B**. A recurrent spiking neural network (RSNN) of the sensorimotor pathway receives synaptic input modeling the sensory stimulation and produces the jaw movement as a behavioral output. **C**. The stimuli and the licking action of the mouse organize the trials into four types (hit, miss, false alarm, and correct rejection). Our goal is to build a model with realistic neural and behavioral variability. Panels A and C are adapted from [34].

deflected and after a 1 s delay period an auditory cue indicates water availability if the mouse licks, whereas in the other 50% of trials (the No-Go trials), there is no whisker deflection and licking after the auditory cue is not rewarded. Throughout the paper we attempt to create a data-constrained model of the six areas that we considered to play a major role in this behavioral task: the primary and secondary whisker somatosensory cortices (wS1, wS2), whisker motor cortices (wM1, wM2), the primary tongue-jaw motor cortex (tjM1) and the anterior lateral motor cortex (ALM), also known as tjM2 (see Figure 1A and 3A). While we focus on this dataset, the method described below aims to be broadly applicable to most contemporary large-scale electrophysiological recordings.

We built a spiking data-constrained model that simulates explicitly a cortical neural network at multiple scales. At the single-cell level, each neuron is either excitatory or inhibitory (the output weights have only positive or negative signs respectively), follows leaky-integrate and fire (LIF) dynamics, and transmits information in the form of spikes with synaptic delays ranging from 2 to 4 ms. At a cortical level, we model six brain areas of the sensory-motor pathway where each area consists of 250 recurrently connected neurons (200 excitatory and 50 inhibitory) as shown in Figure 3A, such that only excitatory neurons project to other areas. Since the jaw movement defines the behavioral output in this task, we also model how the tongue-jaw motor cortices (tjM1, ALM) drive the jaw movements.

Mathematically, we model the spikes $z_{j,k}^t$ of the neuron $j$ at time $t$ in the trial $k$ as a binary number. The spiking dynamics are then driven by the integration of the somatic currents $I_{j,k}^t$ into the membrane voltage $v_{j,k}^t$, by integrating LIF dynamics with a discrete time step $\delta_t = 2$ ms. The jaw movement $y_k^t$ is simulated with a leaky integrator driven by the activity of tjM1 and ALM neurons, followed by an exponential non-linearity. This can be summarized with the following equations, the trial index $k$ is omitted for simplicity:

$$v_j^t = \alpha_j v_j^{t-1} + (1 - \alpha_j) I_j^t - v_{\text{thr},j} z_j^{t-1} + \xi_j^t \tag{1}$$

$$I_j^t = \sum_{d,i} W_{ij}^d z_i^{t-d} + \sum_{d,i} W_{ij}^{\text{in},d} x_i^{t-d} \tag{2}$$

$$\tilde{y}^t = \alpha_{jaw}\tilde{y}^{t-1} + (1 - \alpha_{jaw})\sum_i W_i^{\text{jaw}} z_i^t \tag{3}$$

$$y^t = \exp(\tilde{y}^t) + b \tag{4}$$

where $W_{ij}^d$, $W_{ij}^{\text{in},d}$, $W_i^{\text{jaw}}$, $v_{\text{thr},j}$, and b are model parameters. The membrane time constants $\tau_m = 30$ ms for excitatory and $\tau_m = 10$ ms for inhibitory neurons define $\alpha_j = \exp\left(-\frac{\delta t}{\tau_{m,j}}\right)$ and $\tau_{jaw} = 50$ ms define similarly $\alpha_{jaw}$ which controls the velocity of integration of the membrane voltage and the jaw movement. To implement a soft threshold crossing condition, the spikes inside the recurrent network are sampled with a Bernoulli distribution $z_j^t \sim \mathcal{B}(\sigma(\frac{v_j^t - v_{\text{thr},j}}{v_0}))$, where $v_0$ is the temperature of the sigmoid ($\sigma$). The spike trains $x_i^t$ model the thalamic inputs as simple Poisson neurons producing spikes randomly with a firing probability of 5 Hz and increasing their firing rate when a whisker or auditory stimulation is present (see Appendix A). The last noise source $\xi_j^t$ is an instantaneous Gaussian noise $\xi_j^t$ of standard deviation $\beta v_{\text{thr}}\sqrt{\delta t}$ modeling random inputs from other areas ($\beta$ is a model parameter that is kept constant over time).

**Session stitching** An important aspect of our fitting method is to leverage a dataset of electro-physiological recordings with many sessions. To constrain the neurons in the model to the data, we uniquely assign each neuron in the model to a single neuron from the recordings as illustrated in Figure 2A and 3A. There are 3,810 recorded neurons in the dataset but we take a subset of 250 neurons per area to keep the same number of excitatory and inhibitory neurons in all areas (we were limited by the 290 regular spiking neurons recorded in wS1, and computation resources, mainly, the RAM of the GPU), we ignore the remaining data when fitting the model. This bijective mapping between neurons in the data and the model is fixed throughout the analysis and defines the area and cell types of the neurons in the model. The area is inferred from the location of the corresponding neuron in the dataset and the cell type is inferred from the action potential waveform of the cell (for simplicity, fast-spiking neurons are considered to be inhibitory and regular-spiking neurons as excitatory). Given this assignment, we denote $z_j^{\mathcal{D}}$ as the spike train of neuron $j$ in the dataset and $z_j$ as the spike train of the corresponding neuron in the model; in general, an upper script $\mathcal{D}$ always refers to the recorded data. With this notation the tensor of recorded data $z_j^{\mathcal{D}}$ only exists if neuron $j$ is recorded during session $S$ (denoted $j \in S$), and the activity of neurons not belonging in session $S$ ($j \notin S$) is considered to be unknown. Consequently, two neurons $i$ and $j$ might be synaptically connected in the model although they correspond to neurons recorded in separate sessions. This choice is intended to model network sizes beyond what can be recorded during a single session. Our network is therefore a "collage" of multiple sessions stitched together as illustrated in Figure 2A and 3A. This network is then constrained to the recorded data by optimizing the parameters to minimize the loss functions defined in the following section. Altogether, when modeling the dataset from Esmaeili and colleagues [2], the network consists of 1,500 neurons where each neuron is assigned to one neuron recorded in one of the 28 different recording sessions. Since multiple sessions are typically coming from different animals, we model a "template mouse brain" which is not meant to reflect subject-to-subject differences. In our view, this is coherent with the classical assumption that is made when recording this type of dataset: the features of the circuit being studied are shared among subjects that participate in the experiment. To study differences between sub-groups of subjects (e.g. age group) one could separate the data into two session groups and compare two fitted models. This is beyond the scope of this paper and below we fit a single model to the entire dataset.

## 3 Fitting single-trial variability with the trial matching loss function

We fit the network to the recordings with gradient descent and we rely on surrogate gradients to extend back-propagation to RSNNs [9, 10]. At each iteration until convergence, we simulate a batch of $K = 150$ statistically independent trials. We measure some trial-average and single-trial statistics of the simulated and recorded activity, calculate a loss function, and minimize it with respect to all the trainable parameters of the model via gradient descent and automatic differentiation. This protocol is sometimes referred to as a sample-and-measure method [8] as opposed to the likelihood optimization in GLMs where the network trajectory is clamped to the recorded data during optimization [6]. For the sound evaluation of the model, we separate the recorded trials of every session into a training set (75% of the total trials) and a testing set (25%), that is never used in the fitting procedure. The

split is done in a stratified manner so both sets have the same distribution of trial types. The full optimization lasts for approximately one to three days on a GPU A100-SXM4-40GB.

**Trial-average loss**  We consider the trial-averaged activity over time of each neuron $j$ from every session $\mathcal{T}_{\text{neuron},j}$, sometimes referred also as a neuron's peristimulus time histogram (PSTH). This is defined by $\mathcal{T}_{\text{neuron},j}(\boldsymbol{z}) = \frac{1}{K}\sum_k \boldsymbol{z}_{j,k} * f$ where $f$ is a rolling average filter with a window of 12 ms, and $K$ is the number of trials in a batch of spike trains $\boldsymbol{z}$. The statistics $\mathcal{T}_{\text{neuron},j}(\boldsymbol{z}^{\mathcal{D}})$ are computed similarly on the $K^{\mathcal{D}}$ trials recorded during the session corresponding to neuron $j$. We denote the statistics $\mathcal{T}'_{\text{neuron},j}$ after normalizing each neuron's trial-averaged activity, and we define the trial-averaged loss function as follows:

$$\mathcal{L}_{\text{neuron}} = \sum_j \|\mathcal{T}'_{\text{neuron},j}(\boldsymbol{z}) - \mathcal{T}'_{\text{neuron},j}(\boldsymbol{z}^{\mathcal{D}})\|^2 \; . \tag{5}$$

It is expected from [8] that minimizing this loss function alone generates realistic trial-averaged statistics like the average neuron firing rate.

**Trial matching loss: fitting trial-to-trial variability**  Going beyond trial-averaged statistics, we now describe the *trial matching* loss function to capture the main modes of trial-specific activity. From the previous neuroscience study [2], it appears that population activity in well-chosen areas is characteristic of the trial-specific variability. For instance, intense jaw movements are preceded by increased activity in the tongue-jaw motor cortices, and hit trials are characterized by a secondary transient appearing in the sensory cortices a hundred milliseconds after a whisker stimulation. To define statistics for a single trial $k$ which can capture these features we denote the population-averaged firing rate of an area $A$ as $\mathcal{T}_{A,k}^{S}(\boldsymbol{z}) = \frac{1}{|A\cap S|}\sum_{j\in A\cap S}(\boldsymbol{z}_{j,k} * f)$ where $|A \cap S|$ is the number of neurons in area $A$ recorded in session $S$, the smoothing filter $f$ has a window size of 48 ms and the resulting signal is downsampled to avoid unnecessary redundancy. We write $\mathcal{T}'^{S}_{A,k}$ when each time bin is normalized to mean 0 and standard deviation 1 using the training data from session $S$. We, therefore, use $\mathcal{T}'^{S}_{A,k}(\boldsymbol{z})$ as a feature vector to characterize the recorded area $A$ during each trial $k$ of session $S$, and we build a richer feature vector including all recorded areas and the behavior with the concatenation $\mathcal{T}'^{S}_{\text{trial},k} = (\mathcal{T}'^{S}_{A1,k}, \mathcal{T}'^{S}_{A2,k}, \boldsymbol{y}_k * f)$, where $A1$ and $A2$ are the two areas recorded in session $S$, and $\boldsymbol{y}_k$ is the generated jaw trace from trial $k$. The same feature vector can be computed with the simulated jaw and simulated neural activity by sub-selecting the neurons from areas $A1$ and $A2$ from session $S$ to compare generated and recorded data.

The challenging part is now to define the distance between the recorded statistics $\mathcal{T}'^{S}_{\text{trial},k}(\boldsymbol{z}^{\mathcal{D}})$ and the generated ones $\mathcal{T}'^{S}_{\text{trial},k}(\boldsymbol{z})$. Common choices of distances like the mean square error are not appropriate to compare distributions. This is because the order of trials in a batch of generated/recorded trials has no meaning a priori: there is no reason for the random noise of the first generated trial to correspond to the first recorded trial – rather we want to compare unordered sets of trials and penalize if any generated trial is very far from any recorded trial.

Formalizing this mathematically we consider a distance between distributions inspired by the optimal transport literature. Since the plain mean-squared error cannot be used, we use the mean-squared error of the optimal assignment between pairs of recorded and generated trials: we select randomly $K' = \min(K, K^{\mathcal{D}})$ generated and recorded trials ($K$ and $K^{\mathcal{D}}$ are respectively the number of generated and recorded trials in one session), and this optimal assignment is formalized by the integer permutation $\pi : \{1, \dots K'\} \to \{1, \dots K'\}$. Then using the feature vector $\mathcal{T}'^{S}_{\text{trial},k}$ for any trial $k$, we define the hard *trial matching* loss function as follows by the sum $\mathcal{L}_{\text{trial}} = \sum_S \mathcal{L}_{\text{trial}}^{S}$ with:

$$\mathcal{L}_{\text{trial}}^{S} = \min_{\pi} \sum_k ||\mathcal{T}'^{S}_{\text{trial},k}(\boldsymbol{z}) - \mathcal{T}'^{S}_{\text{trial},\pi(k)}(\boldsymbol{z}^{\mathcal{D}})||^2 \; . \tag{6}$$

We compute this loss function identically on all the recorded sessions and take the summed gradients to update the parameters. Each evaluation of this loss function involves the computation of the optimal trial assignment $\pi$ which can be computed with the Hungarian algorithm [35] (see linear_sum_assignment for an implementation in scipy). This is not the only way to define a distance between distributions of statistics $\mathcal{T}'^{S}_{\text{trial},k}$. In fact, this choice poses a potential problem because the optimization over $\pi$ is a discrete optimization problem, so we have to assume that $\pi$ is a constant with respect to the parameters when computing the loss gradients. We also tested alternative

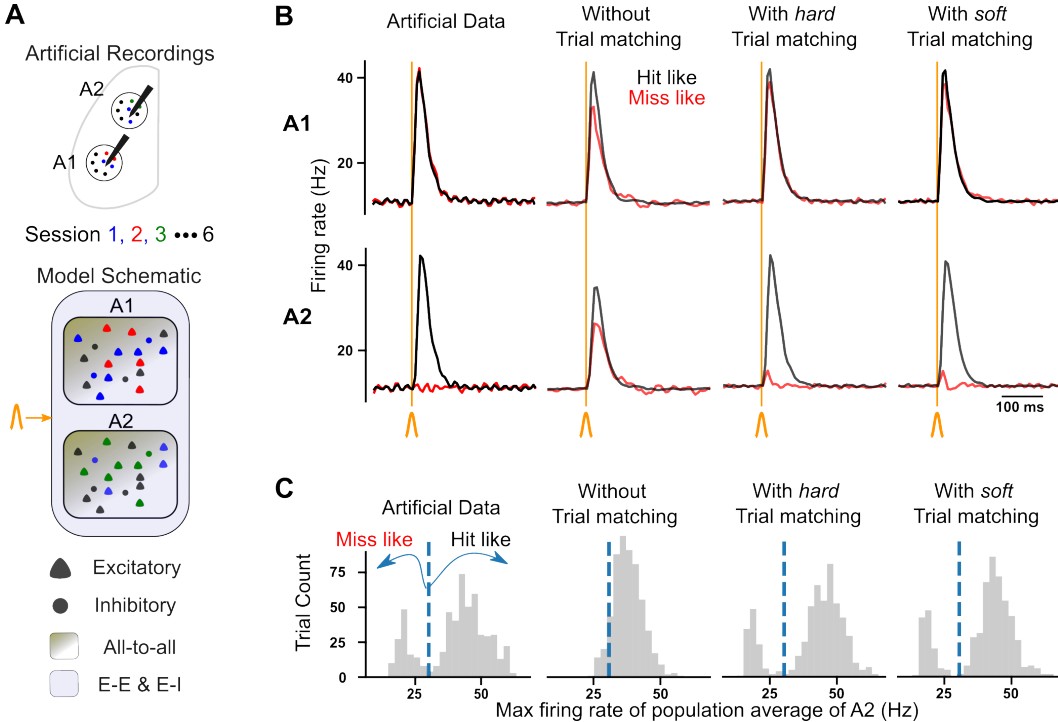

Figure 2: **Artificial Dataset.** **A**. Session stitching: every neuron from the recordings is uniquely mapped to a neuron from our model. For example, an excitatory neuron from our model that belongs in the putative A1 is mapped to an excitatory neuron "recorded" in A1. In our network, we constrain the connectivity so that only excitatory neurons can project across different brain regions. **B**. The first area (A1) responds equally in a hit-like and a miss-like trial, while the second area (A2) responds only in hit-like trials. A model that does not use trial matching cannot capture the bimodal distribution of A2. **C**. Distribution of max firing rate of the population average of A2 from each trial. Only the trial matching algorithms retrieve the bimodal behavior of A2.

choices relying on a relaxation of the hard assignment into a smooth and differentiable bi-stochastic matrix. This results in the soft *trial matching* loss function, which replaces the optimization over $\pi$ by the Sinkhorn divergence [12, 13] (see the `geomloss` package for implementation in pytorch [13]). In practice, to minimize both $\mathcal{L}_{\text{trial}}$ (either the soft or hard version) and $\mathcal{L}_{\text{neuron}}$ simultaneously we optimize them in an additive fashion with a parameter-free multi-task method from deep learning which re-weights the two loss functions to ensure that their gradients have comparable scales (see [36] for a similar implementation).

## 4 Simulation results

**Validation using an artificial dataset** We generated an artificial dataset with two distinct areas with 250 neurons each to showcase the effect of trial variability. In this dataset A1 (representing a sensory area) always responds to a stimulus while A2 (representing a motor area) responds to the stimulus in only $80\%$ of the trials, see Figure 2. This is a toy representation of the variability that is observed in the real data recorded in mice, so we construct the artificial data so that a recording resembles a hit trial ("hit-like") if the transient activity in A2 is higher than 30 Hz (otherwise it's a "miss-like" trial). From the results of our simulations Figure 2B-C, we can observe that the models that use trial matching (either soft *trial matching* or hard *trial matching*) can re-generate faithfully this bimodal response distribution ("hit-like" and "miss-like") in A2. In this dataset we saw little difference between the solutions of soft and hard *trial matching*, if any, soft *trial matching* reached its optimal performance with fewer iterations (see Figure A.2C). As expected, when the model is only trained to minimize the neuron loss for trial-averaged statistics, it cannot generate stochastically this bimodal distribution and consistently generates instead a noisy average response.

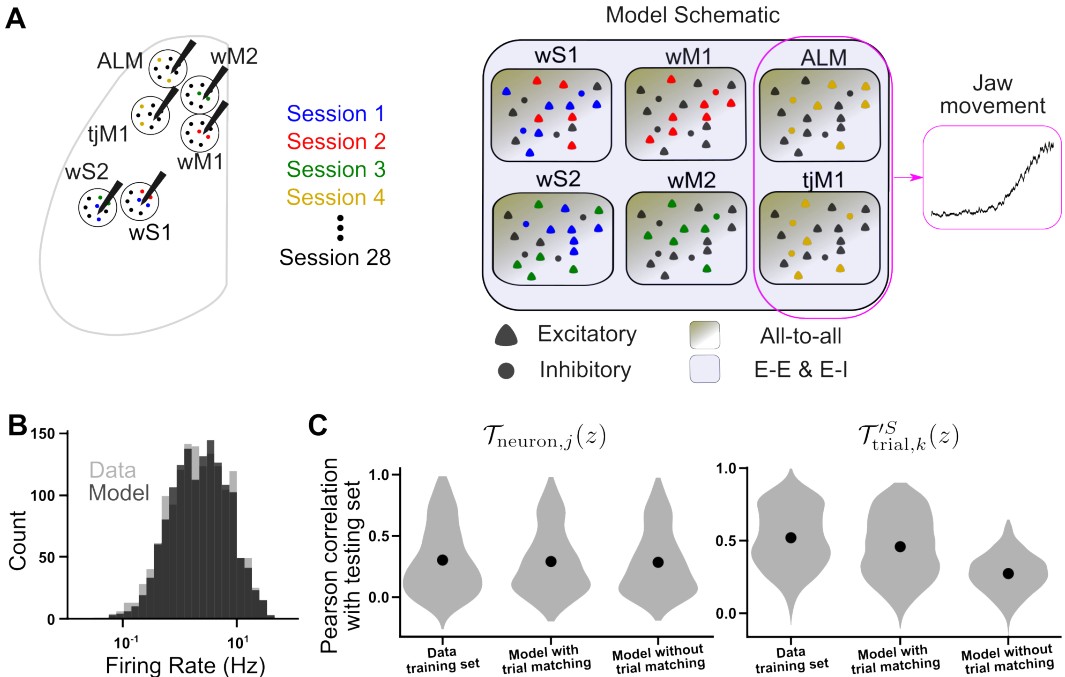

Figure 3: **Large-scale electrophysiology recordings dataset. A**. Session stitching: Every neuron from the recordings across sessions is uniquely mapped to a neuron from our model. For example, an excitatory neuron from our model that belongs in the putative area tjM1 is mapped to an excitatory neuron recorded from tjM1. In the pink box are the areas from which we decode the jaw trace. **B**. Baseline firing rate histogram, 200 ms before the whisker stimulus, from each neuron of our model and the recordings. **C**. Left: Pearson correlation of the PSTH, the violin plots represent the Pearson correlations across neurons. Right: *trial-matched Pearson correlation* of $\mathcal{T}'^{S}_{\mathrm{trial},k}$, the violin plots represent the distribution over 200 generated and recorded trial pairs.

**Delayed whisker tactile detection dataset**    We then apply our modeling approach to the real large-scale electrophysiology recordings from [2]. After optimization, we verify quantitatively that our model generates activity that is similar to the recordings in terms of trial-averaged statistics.

First, we see that the 1,500 neurons in the network exhibit a realistic diversity of averaged firing rates: the distribution of neuron firing rates is log-normal and matches closely the distribution extracted from the data in Figure 3B. Second, the single-neuron PSTHs of our model are a close match to the PSTHs from the recordings. This can be quantified by the Pearson trial-averaged correlation between the generated and held-out test trials which we did not use for parameter fitting. We obtain an averaged Pearson correlation of $0.30 \pm 0.01$ which is very close to the Pearson correlation obtained when comparing the training and testing sets $0.31 \pm 0.01$. Figure 3C shows how the trial-averaged correlation is distributed over neurons. As expected, this trial averaged metric is not affected if we do not use *trial matching* ($0.30 \pm 0.01$).

To quantify how the models capture the trial-to-trial variability, we then quantify how the distributions of neural activity and jaw movement are consistent between data and model. So we need to define the *trial-matched Pearson correlation* to compute a Pearson correlation between the distribution of trial statistics $\mathcal{T}'^{S}_{\mathrm{trial},k}$ which are unordered sets of trials. So we compute the optimal assignment $\pi$ between trial pairs from the data and the recordings, and we report the averaged Pearson correlation overall trial pairs. Between the data and the model, we measure a *trial-matched Pearson correlation* of $0.48 \pm 0.01$, with a performance ceiling at $0.52 \pm 0.01$ obtained by comparing the training and testing set directly (see Figure 3C for details). For reference, the model without *trial matching* has a lower *trial-matched Pearson correlation* $0.28 \pm 0.003$.

**Successful recovery of trial type distribution**    While the neuronal activity is recorded, the behavioral response of the animal is also variable. When mice receive a stimulation they perform correctly

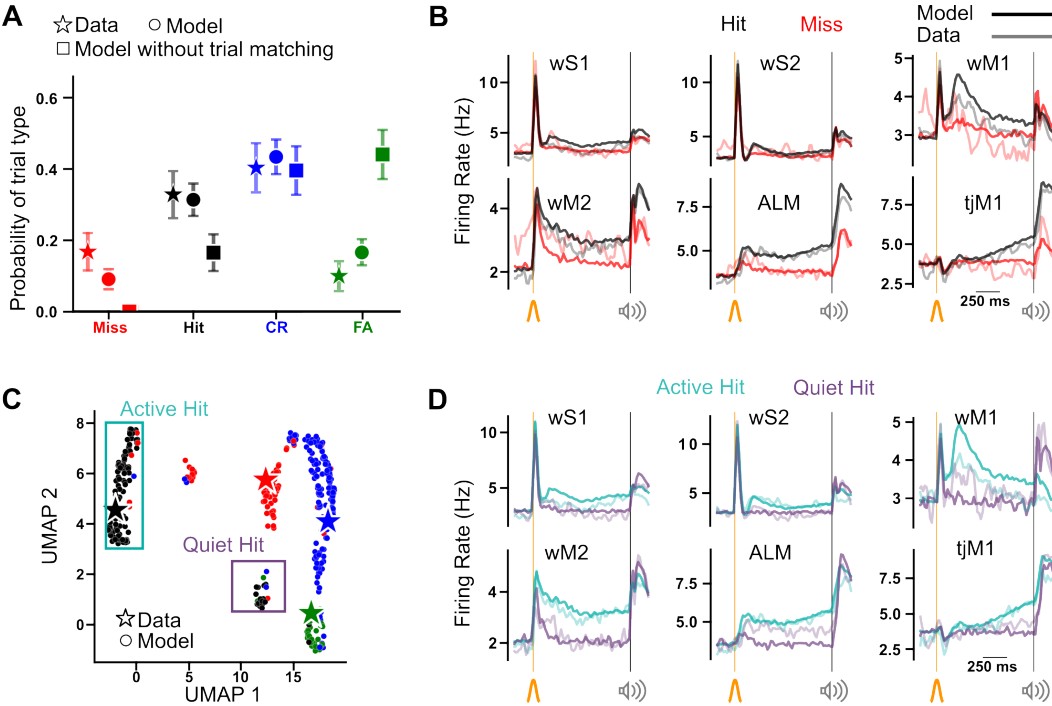

Figure 4: **Emergent trial types. A**. Trial type distribution from the recordings and the models. The whiskers show the 95% confidence interval of the trial type frequency. In this panel, trial types use the template matching method to avoid disadvantaging models without *trial matching* which have almost no variability in the jaw movement, see Figure A.3. **B**. Population and trial-averaged neuronal activity per area and per Hit and Miss trial type from the 400 simulated trials from the model against the averaged recordings from the testing set. **C**. Two-dimensional UMAP representation of the $\mathcal{T}_{\text{trial},k}$ of 400 simulated trials. The jaw movement is not used in this representation. **D**. For the model, we separated the active hit and quiet hit trials based on their location in the UMAP. For the data, we separated the active hit and quiet hit trials based on the jaw movement as in [2].

with a 66% hit rate, while in the absence of a stimulus, mice still falsely lick with a 20% false alarm rate. Even in correct trials, the neural activity reflects variability which is correlated to uninstructed jaw and tongue movements [2].

We evaluate the distribution of trial types (hit, miss, correct rejection, and false alarm) from our fitted network model. Indeed, the 95% confidence intervals of the estimated trial type frequencies are always overlapping between the model and the data (see Figure 4A). In this Figure, we classify the trial type with a nearest-neighbor-like classifier using only the neural activity (see Figure A.3A). In contrast, a model without the *trial matching* would fail completely because it always produces averaged trajectories instead of capturing the multi-modal variability of the data as seen in Figure 4A. With *trial matching* it is even possible to classify trial types using jaw movement. To define equivalent trial types in the model, we rely on the presence or absence of the stimulation and a classifier to identify a lick action given the jaw movements. This classifier is a multi-layer perceptron trained to predict a lick action on the water dispenser given the recorded jaw movements (like in the data, the model can "move" the jaw without inducing a lick action). After optimization with *trial matching*, since the jaw movement $y_k^t$ is contained in the fitted statistics $\mathcal{T}_{\text{trial},k}^S$, the distribution of jaw movement and the trial types are similar in the fitted model and the trial type distribution remains consistent. In Figure 4B we show population-averaged activity traces where the jaw is used to determine the trial type.

**Unsupervised discovery of modes of variability** So far we have analyzed whether the variability among the main four trial types was expressed in the model, but the existence of these four trial types is not enforced explicitly in the loss function. Rather, the *trial matching* loss function aims to match the overall statistics of the distributions and it has discovered these four main modes of trial-

variability without explicit supervision. A consequence is that our model has possibly generated other modes of variability which are needed for the model to explain the full distribution of recorded data.

To summarize the full distribution of trials, we project the neural activity in a 2D space in Figure 4C. This representation recapitulates the modes of co-variability in all six areas and is not directly possible to be generated from the dataset because trials from distinct sessions have non-overlapping recorded units. This representation becomes possible with a generative model like ours that can capture the distribution of trial-to-trial variability from incomplete recordings. Technically, we generate $400$ trials and we apply UMAP to the concatenation: $(\mathcal{T}_{wS1,k}, \dots \mathcal{T}_{ALM,k})$ where each vector $\mathcal{T}_{A,k}$ includes now all neurons simulated in area $A$ so the generated vectors contain data about all sessions and areas. To confirm that the generated representation of the distribution is consistent with the data we display template vectors for each trial condition $c$ that are calculated directly on the recorded data trials. These templates are drawn with stars in Figure 4C and are computed as follows: the feature $\mathcal{T}_{A,c}^{\mathcal{D}}$ of this template vector is computed by averaging the population activity of area $A$ in all recorded trials from all sessions (see Appendix C for details), the averaged vectors are then concatenated and projected in the 2D UMAP space.

The consistency between the condition-averaged templates and the full trial distribution is confirmed visually in Figure 4C. The emerging distribution in this visualization is grouped in clusters. We observe that the template vectors representing the correct rejection, miss, and false alarm trials are located at the center of the corresponding cluster of generated trials. It also appears that the generated trial distribution provides richer information about the trial-to-trial variability that is hard to study using condition-averaged templates. For instance, we were curious why the generated hit trials are split into two clusters (see the two boxed clusters in Figure 4C). Looking more closely at the difference between those two clusters of generated trials, it turned out that their difference can be explained by a simple feature: 85% of the generated hit trials on the left-hand cluster of panel 4C have intense jaw movements during the delay period ($\max_t |y^t - y^{t-1}| > 4\delta$ where $\delta$ is the standard deviation of $|y^t - y^{t-1}|$ in the 200 ms before whisker stimulation). This criterion happens to be similar to the one used by experimentalists [2] to separate the hit trials in the recorded data, so we also refer to them as the "active hit" and "quiet hit" trials and show the population activity in Figure 4D. This interesting correspondence highlights the potential for our method to study the structure of the trial-to-trial variability without a priori assumptions of the trial condition partitions. We conclude that our modeling approach can be used for a hypothesis-free identification of modes of trial-to-trial variability, even when they reflect task-irrelevant behavior.

## 5   Discussion

We introduced a generative modeling approach where a data-constrained RSNN is fitted to multi-session electrophysiology data. The two major innovations of this paper are (1) the technical progress towards multi-session RSNN fitted with automatic differentiation, and (2) a *trial matching* loss function to match the trial-to-trial variability in recorded and generated data.

**Interpretable mechanistic model of activity and behavior**   Our model has a radically different objective in comparison with other deep-learning models: our RSNN is aimed to be biophysically interpretable. In Figure A.5 in the appendix, we illustrate a strong difference between our interpretable RSNN model and the more generic LFADS method which generates neural network activity with deep learning models. The most striking difference is that LFADS [22] relies on Gated recurrent units (GRU) which are abstract recurrent models for which there is no explicit mapping with real neurons. In our case, by construction there is a one-to-one mapping between an RSNN spiking unit and a recorded neuron. We further illustrated a consequence of our model in Figure A.5, we designed an extension of the artificial dataset where two putative areas are activated with a substantial delay of 20, or 200 ms. In real recordings, this would be only possible if unrecorded areas were maintaining some network activity during this long delay, but there is no plausible network hypothesis with two areas that could solve the task. When the delay between the areas is plausibly short (20 ms), the generated data from our RSNN model and LFADS are similarly accurate. For longer delays, the constraint in the RSNN model cannot allow for an implausible network solution by design so our optimization collapses and does not provide any speculative circuit hypothesis for this artificial dataset unless we allow our network to have implausible long synaptic transmission delays,

see Figure A.5B-C. In contrast, LFADS can generate solutions independently of the delay duration, partly thanks to its bidirectional GRU units with efficient internal memory gating which can handle very large delays. In the long term, we hope that the interpretability of our method will be able to capture biological mechanisms (e.g. predicting network structure, causal interactions between areas, and anatomical connectivity), but in this paper, we have focused on numerical and methodological questions which are getting us one step closer to this long-term objective.

**Mechanisms of latent dynamics**    A long-standing debate is whether the brain computes with low-dimensional latent representations and how that is implemented in a neural circuit. Another reason for the LFADS to fit implausibly long delays, see Figure A.5, is inherent to the deep auto-encoder setting. By construction, LFADS generates trial-to-trial variability from low-dimensional latent representations, so the variability is sourced by the latent variable which contains all the trial-specific information. This is in stark contrast with our approach, where we see the emergence of structured manifolds in the trial-to-trial variability of the RSNN from the noise source of the network dynamics. In the UMAP representation of Figure 4C, we visualize a low-dimensional manifold of trial distribution although we did not enforce explicitly the presence of low-dimensional latent dynamics. Structure in the trial-to-trial variability emerges because the RSNN is capable of transforming the unstructured noise sources (stochastic spikes and Gaussian input current) into a low-dimensional trial-to-trial variability – a typical variational auto-encoder setting would not achieve this.

Note that it is also possible to add a random low-dimensional latent as a source of low-dimensional variability in our RSNN as it is done with LFADS. In the Figure A.4, we reproduce our results on the multi-session dataset from [2] while assuming that all voltages $v_{i,k}^t$ have a trial-specific excitability offset $\xi_{i,k}$ using a 5-dimensional gaussian noise $\psi_k$ and a one-hidden-layer perceptron $F_\theta$ such that $\xi_{i,k} = F_{\theta,i}(\psi_k)$. The latent variable, therefore, implements a neuron-specific excitability offset coming from an unknown source. Unsurprisingly this latent noise model accelerates drastically the optimization, probably because $\xi_{i,k}$ is an efficient noise source for minimizing $\mathcal{L}_{\text{trial}}$. More remarkably, the final fitting performance metrics are the same with and without the low-dimensional drive. It suggests that the extra assumption of a low-dimensional input from an unknown source is convenient but not necessary to generate realistic variability. Our first hypothesis that the recorded circuit alone produces the distribution of possible activity and behavior is sufficient. We speculate that providing this low-dimensional input to the model is sometimes counterproductive if the end goal is to identify the mechanism by which the circuit produces the low-dimensional dynamics.

**Alternative loss functions to capture variability**    In our optimization, the trial-matching loss function constrains the network to produce realistic trial-to-trial variability. The main alternative methods to constrain the trial-to-trial variability would be likelihood-based approaches [6, 15] or spike-GANs [27, 33]. These methods are appealing as they do not depend on the choice of trial statistics $\mathcal{T}_{\text{trial},k}^S$. Since these methods were never applied with a multi-session data-constrained RSNN we explored how to extend them to our setting and compare the results. We tested these alternatives on the Artificial dataset in Figure A.2. The likelihood of the recorded spike trains [6, 15] cannot be defined with multiple sessions because we cannot clamp neurons that are not recorded (see [8] for details). The closest implementation that we could consider was to let the network simulate the data "freely" which requires, therefore, an optimal assignment between recorded and generated data, so it is a form of *trial-matched likelihood*). With this loss function, we could not retrieve the bi-model hit versus miss trial type distribution unless it is optimized jointly with $\mathcal{L}_{\text{trial}}$.

We also tested the implementation of a spike-GAN discriminator. In GANs the min-max optimization is notoriously hard to tune, and we were unable to train our generator with a generic spike-GAN discriminator from scratch (probably because the biological constraints of our generator affect the robustness of the optimization). In our hands, it only worked when the GAN discriminator was fed directly with the trial statistics $\mathcal{T}_{\text{trial},k}^S$ and the network was jointly fitted to the trial-averaged loss $\mathcal{L}_{\text{neuron}}$. It shows that a GAN objective and the *trial matching* loss function hold a similar role. We conclude that both of these clamping-free methods are promising to fit data-constrained RSNNs. What differs between them, however, is that *trial matching* replaces the discriminator with the optimal assignment $\pi$ and the statistics $\mathcal{T}$ which are parameter-free, making them easy to use and numerically robust. It is conceivable for future work the best results are obtained by combining *trial matching* with other GAN-like generative methods.

## Acknowledgments and Disclosure of Funding

This research was supported by the Swiss National Science Foundation (no. 31003A_182010, TMAG-3_209271, 200020_207426), and Sinergia Project CRSII5_198612. Many thanks to Lénaïc Chizat, Vivien Seguy, James Isbister, Shuqi Wang and Vahid Esmaeili for their helpful discussions.

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
