# Appendix

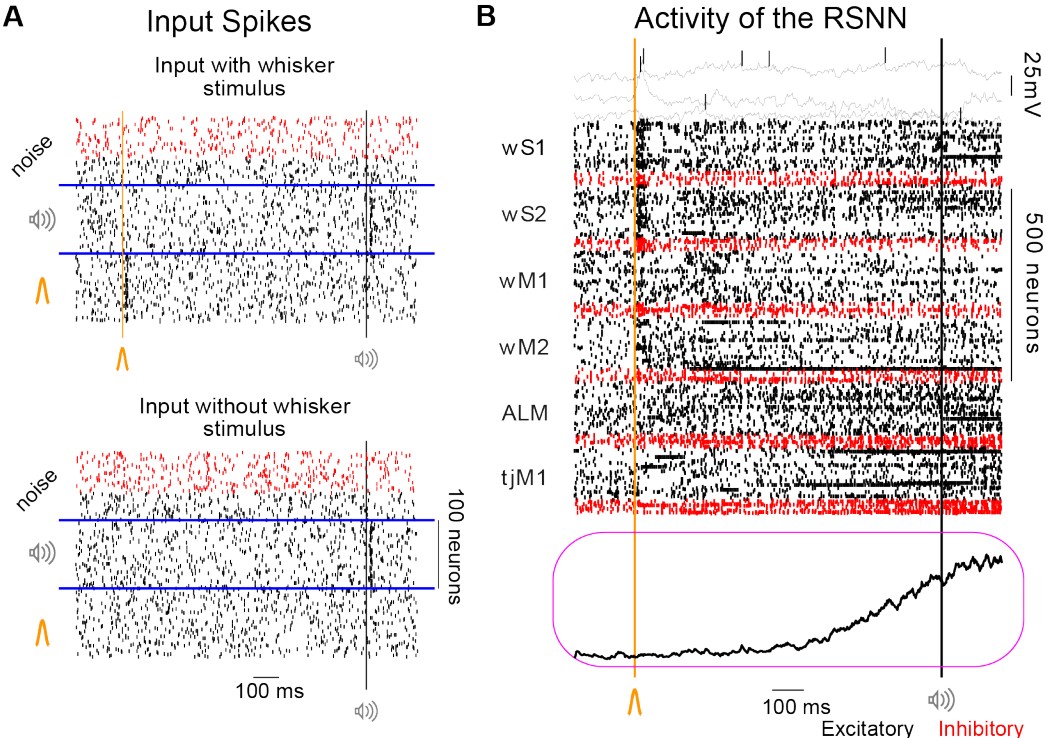

Figure A.1: **Input spikes. A**. The input spikes, $x_i^t$, are one of the main drivers of the activity of our RSNN. They are 300 Poisson neurons, where the first 100 encode the whisker stimulus, the next 100 encode the auditory cue and the last 100 act as an extra noise source for our model. Out of the 300 neurons, 60 of them are inhibitory (red). The input neurons project unrestrictedly to the whole RSNN. The baseline firing rate of all input neurons is 5 Hz. The whisker stimulus and auditory cue are encoded with an increase of the firing rate for 10 ms, starting 4 ms after the onset of the actual stimuli. **B**. Activity of the RSNN for one hit trial. At the top, are voltage traces from 3 neurons, below the spiking activity of the whole model, and at the bottom is the decoded jaw movement for this trial.

## A   Input Spikes

Some input neurons model thalamic input or random cortical input noise. Practically these input neurons are simulated with 300 Poisson neurons as seen in Figure A.1. Two-thirds are encoding the thalamic sensory inputs and the remaining can be interpreted as random cortical neurons. All these neurons have a background firing activity of 5 Hz, but the 200 thalamic neurons, have a sharp and phasic increase to 20-40 Hz for a 10 ms window when the stimuli are presented. Precisely the increase in the activity starts at 4 ms after the actual stimulation, to take into account the time that the tactile/auditory signal needs to reach the thalamus. The remaining Poisson neurons are mostly inhibitory and help the network balance its activity since all other input spikes are excitatory. Figure A.1A shows the input spikes for two trials one with and one without a whisker stimulus.

## B   Alternative methods to fit trial variability

The two main alternative methods to constrain the trial variability of a generative model are likelihood-based approaches and the spike-GAN. In theory, these two methods can fit higher-order statistics of the data so they should be able to capture trial-to-trial variability. As explained below, in practice we did not see that they could easily replace the benefits of the trial matching loss functions.

In order to make a comparison of these two methods with *trial matching* we tested them on the artificial data, see Figure A.2.

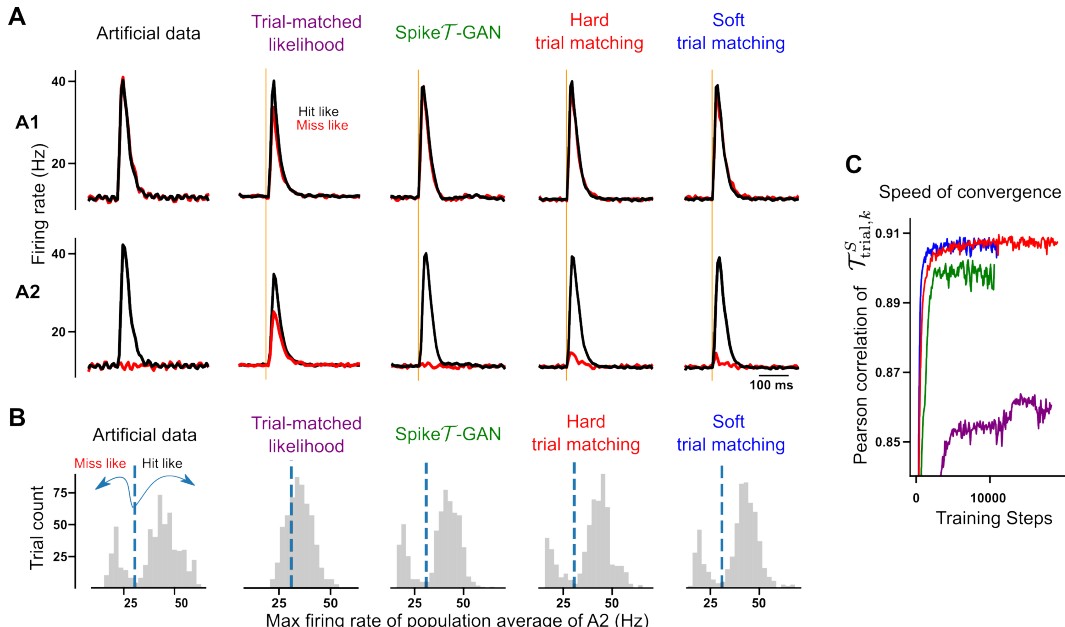

Figure A.2: **Fitting trial variability.** **A**. The first area (A1) responds equally in a hit-like and a miss-like trial, while the second area (A2) responds only in hit-like trials. All models, except the *trial-matched likelihood*, behave similarly and they can capture the bimodal distribution of A2. **B**. Distribution of max firing rate of the population average of A2 from each trial. **C**. Evolution of the *trial-matched Pearson correlation* of our model during the training. We stop the simulations if there was no improvement in the loss function after 8000 training steps and we show the first 18000 steps for visualization purposes.

For the **likelihood-based approaches**, we made two main adjustments to be applicable to our setting. First, we did not "clamp" the neural activity during training meaning that when the network evaluated timestep $t + 1$ the activity was not fixed to the recordings from timestep $t$. This was necessary in our case because we fitted multiple sessions. Second, we optimally assigned the simulated trials with the recorded trials as in the *trial matching* loss, $\mathcal{L}_{\text{trial}}$. This resulted in the following loss function:

$$\mathcal{L} = \min_{\pi} BCE(\boldsymbol{z}_{j,k}, \boldsymbol{z}^{\mathcal{D}}_{j,\pi(k)}) \,, \tag{1}$$

where BCE is the binary cross-entropy loss. We call this method *trial-matched likelihood*.

For the **spike-GAN**, the training of a generic spike-train classifier as a discriminator was not successful in our hands. We speculate that there are two main reasons for this. First, the data are very noisy from trial to trial, and even first-order statistics sometimes are not consistent, see Figure 3C. Second, our generator is a recurrent spiking neural network that is more difficult to train than the deep-learning spike-train generators used previously. We could make it work however with a modified discriminator, which does not receive the full spike trains but only the $\mathcal{T}^S_{\text{trial},k}$ statistics. We call this variant spike$\mathcal{T}$-GAN. Since spike$\mathcal{T}$-GAN has access only to the $\mathcal{T}^S_{\text{trial},k}$ we minimize the GAN loss function jointly with the $\mathcal{L}_{\text{neuron}}$ loss.

**Comparison results**    As shown in Figure A.2, the *trial-matched likelihood* implementation fails to reconstruct the bimodal distribution of the data, while the spike$\mathcal{T}$-GAN reaches similar accuracy and behavior as our *trial matching* methods. In Figure A.2B we also observed that the spike$\mathcal{T}$-GAN finds the correct bi-modal distribution with roughly the correct proportion of "hit-like" and "miss-like" trials. In this sense, spike$\mathcal{T}$-GAN loss function and the *trial matching* loss functions have a similar function.

However despite the simplicity of the discriminator which already receives population average statistics, both *hard* and *soft trial matching* converge much faster than the spike$\mathcal{T}$-GAN. It happens be-

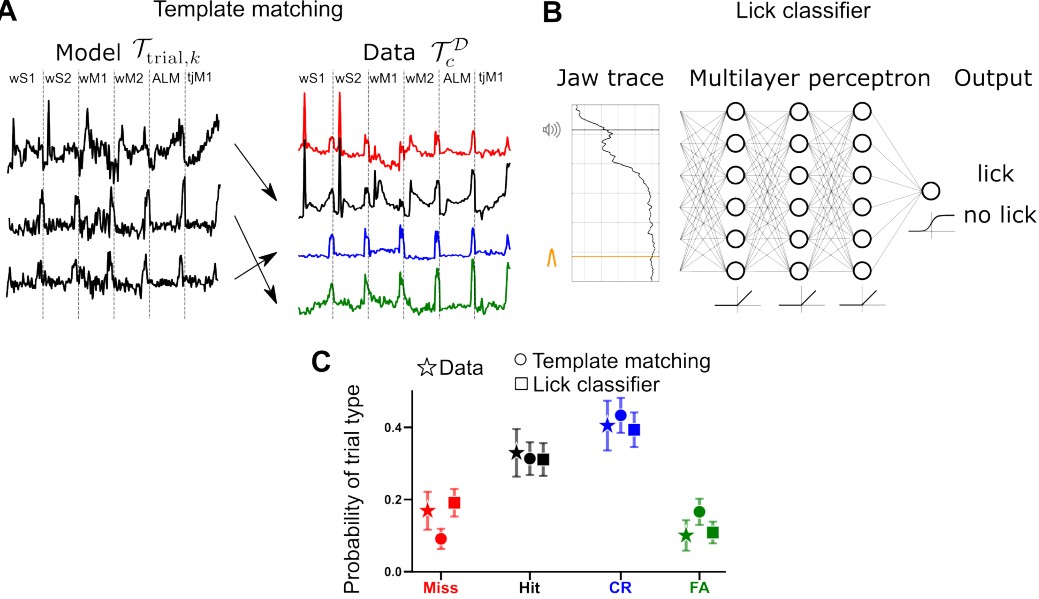

Figure A.3: **Trial type classification. A**. Template matching or nearest-neighbor-like classifier. We calculate the per area and per trial type PSTH from the recordings across all sessions, $\mathcal{T}_{A,c}^{\mathcal{D}}$. We concatenate these vectors producing a single vector per trial condition $\mathcal{T}_c^{\mathcal{D}} = (\mathcal{T}_{wS1,c}^{\mathcal{D}} \ldots \mathcal{T}_{tjM1,c}^{\mathcal{D}})$. We call these signals templates since they fully describe the neuronal activity of each trial type. Thereafter we calculate an equivalent signal for each trial from our model, which is effectively the $\mathcal{T}_{\mathrm{trial},k}$, we find which is the closest template vector. **B**. Multilayer perceptron (MLP) lick classifier. We use a 3 hidden layer MLP classifier to detect if a trial was a lick trial. As an input, we use the filtered and binned jaw movement. Each hidden layer has 128 units and a RELU activation function. The output neuron has a sigmoid activation function. We train the network with jaw movements from the training set, where we know for each trial the behavioral outcome (lick vs no lick). We use the Binary Cross Entropy loss function and optimize the network using the ADAM optimizer. **C**. Both trial-type classifiers verify that our model produces the same trial-type distribution as the recordings.

cause the discriminator needs to be trained along with the generator in GANs, while the comparable competitive optimization is implemented at each iteration by $\pi$ in *trial matching*. We also see that the *soft* version converges faster than the *hard*, either because the computation of $\pi$ is taken into account in the auto-differentiation, or because it implicitly regularizes the distributions which can be favorable if $\mathcal{T}$ has high dimensions [13].

## C   Trial type classification

In the context of studying behavioral outcomes and neuronal activity, trial-type classification plays a crucial role in understanding the underlying processes. This section describes two approaches employed for trial type classification: template matching (nearest-neighborhood-like classifier) and Multilayer Perceptron (MLP) lick classifier.

**Template Matching or Nearest-Neighbor-Like Classifier**    We calculate the per-area and per-trial type peristimulus time histogram (PSTH) using the recordings across all experimental sessions, denoted as $\mathcal{T}_{A,c}^{\mathcal{D}}$. By concatenating these vectors, we obtain a single vector per trial condition, denoted as $\mathcal{T}_c^{\mathcal{D}} = (\mathcal{T}_{wS1,c}^{\mathcal{D}} \ldots \mathcal{T}_{tjM1,c}^{\mathcal{D}})$. These vectors serve as templates since they provide a comprehensive description of the neuronal activity for each trial type. Next, we generate an equivalent signal, $\mathcal{T}_{\mathrm{trial},k}$, for each trial $k$ using our model. By comparing this signal to the template vectors, we determine which trial type is the closest. This trial type classification method, which we refer to as template matching or nearest-neighbor-like classifier, enables us to identify the trial type based only

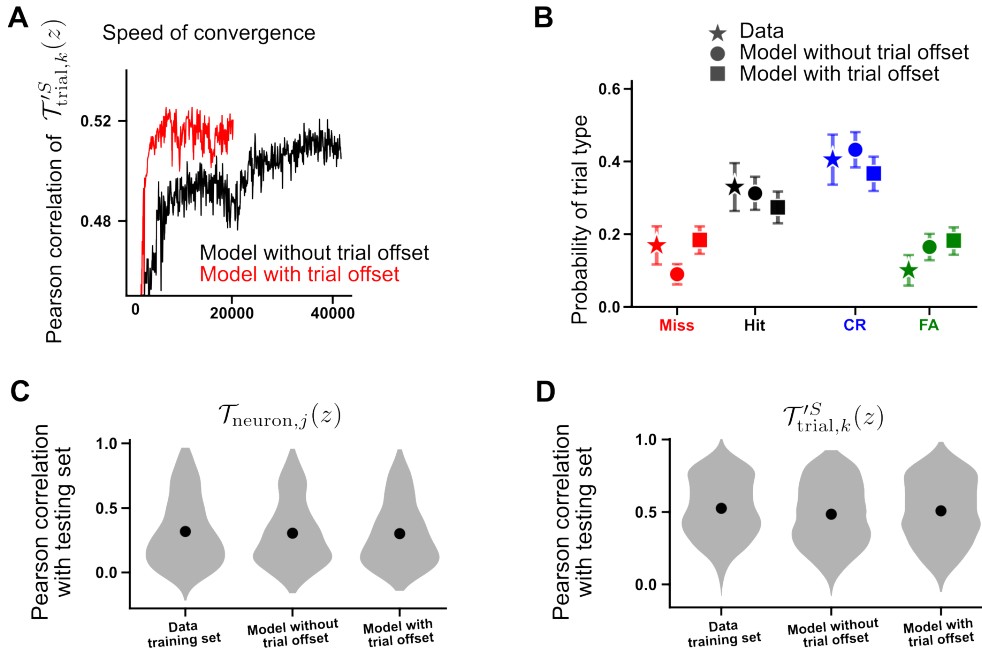

Figure A.4: **Trial-specific offset.** **A**. Evolution of the *trial-matched Pearson correlation* of our model during the training. We stop the simulations if there was no improvement in the loss function after 8000 training steps. **B**. Trial type distribution from the recordings, the model used for section 4, and a model with trial-specific offset. The whiskers show the $95\%$ confidence interval, assuming that each trial type has a Bernoulli distribution. Here we classify trial types using the template matching method. **C**. Pearson correlation of the $\mathcal{T}_{\mathrm{neuron},j}$, the violin plots represent the Pearson correlations across neurons. **D**. *trial-matched Pearson correlation* of $\mathcal{T}'^{S}_{trial,k}$, the violin plots represent the distribution over 200 generated and recorded trial pairs.

on the model's neural activity, see Figure A.3A. The vectors described here are the ones that were used for the UMAP dimensionality reduction in Figure 4C.

**Multilayer Perceptron (MLP) Lick Classifier** We utilize a multilayer perceptron neural network to detect whether a trial corresponds to a lick or no-lick event. The input is the filtered and binned jaw movement. The classifier consists of three hidden layers, each comprising 128 hidden units. The Rectified Linear Unit (ReLU) is chosen as the activation function for the hidden layers, and the sigmoid function for the output layer, Figure A.3B shows the network architecture. To optimize the MLP lick classifier, we use the Binary Cross Entropy loss function and with the ADAM optimizer we iteratively update the network's weights and biases, minimizing the loss function. After the training process, the MLP lick classifier achieves 93.5% correct classification on the trials of the testing set from the recordings and it can be used directly to classify the model-generated jaw movements.

In Figure A.3C, we can see that both the template matching and MLP lick classifiers reproduce the trial-type distribution observed in the recordings.

## D    Comparing our model with a deep-learning generative model

In order to compare our model with LFADS, we generated a variant of the previous artificial dataset. The main difference with the artificial dataset from Figure 2 is that the putative motor area A2 has a bump of activity that arrives with a variable delay after the bump of activity in area A1. We consider two settings, either the delay is 20ms which is a realistic delay between two connected cortical areas, or this delay is much larger 200ms which is implausible. In a real brain, this would only be expected if other unrecorded areas are holding some form of working memory during the delay. Here, since the model is only modeling the recorded data, this example is designed to be incompatible with the model assumptions.

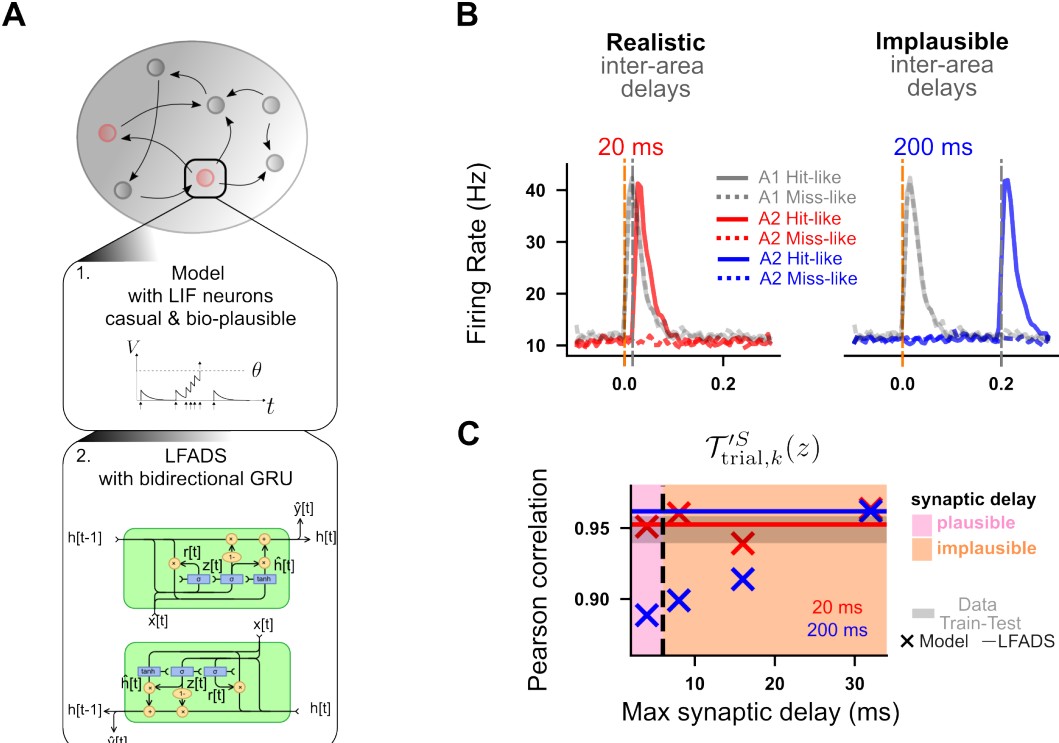

Figure A.5: **Similar accuracy with a deep learning generative model (LFADS) in the bioplausible regime. A**. Schematic representation of an RNN where in 1. every neuron is simulated as leaky integrate and fire neuron with casual dynamics, and in 2. the main nodes of the RNN are biderectional Gated Recurrent Units (GRU) and the model is simulated forward and backward in time. **B**. Conditional trial averaged PSTHs for areas A1 and A2 from three artificial datasets. The first area (A1) responds equally in hit-like and miss-like trials, while the second area (A2) in all datasets responds only 80% of the time, with delays of 20, and 200 ms. **C**. *Trial-matched Pearson correlation* ($\mathcal{T}'^{,S}_{\text{trial},k}$) of different RSNN models with only change in the synaptic transmission delay. The solid lines show the $\mathcal{T}'^{,S}_{\text{trial},k}$ of LFADS. The gray line is the $\mathcal{T}'^{,S}_{\text{trial},k}$ between the training and test set in the three artificial datasets. The dashed line indicates that a biologically plausible synaptic transmission delay is below 6 ms. The GRU diagram in Panel A is adapted from a tutorial on RNNs.

When the delay corresponds is realistic (20 ms) the model performed on par with LFADS, we show the results in Figure A.5C (we implemented LFADS using the tutorial https://github.com/snel-repo/lfads-tutorial and default hyper-parameters in most cases). When area A2 arrives with an implausible delay (200 ms) the Pearson correlation drops drastically. We could however recover a high performance when increasing the synaptic delay to an implausible range (around 30ms).

We conclude that an interpretable model like ours can perform on par with powerfull methods like LFADS when there exists a plausible model hypothesis. In contrast, in a method like LFADS, since there is no explicit causal or biophysical interpretability that is explicitly required, the model is also capable of generating problematic data.