# OpenReview forum: "Trial matching: capturing variability with data-constrained spiking neural networks"
_NeurIPS.cc/2023/Conference — NeurIPS 2023 poster_

### Official Review · Reviewer_Cbej · 2023-07-06

**Soundness:** 3 good
**Presentation:** 3 good
**Contribution:** 3 good
**Rating:** 7
**Confidence:** 4

**Summary:**

This paper considers fitting high-fidelity recurrent spiking neural networks to multiple populations of neurons across sessions. The networks include LIF neurons for each fitted neuron and model within & across region interactions at fine timescales, and include a soft-thresholding mechanism for neural spiking. The model is fit to match the responses of the recorded neurons both across trials and to generate variability across trials that is consistent with the variability observed in the data. For the latter, the authors propose an optimal-transport based loss on the distributions of sampled recorded trials and trials simulated from the model.

The authors validate the proposed method in simulation, showing that the trial matching term in the objective is critical for capturing variability across trial types in neural data. Then, they apply it to a large multi-population neural dataset, showing the model captures trends across trial types in this dataset as well.

**Strengths:**

The motivation and methods are clear and novel, and the resulting model appears to be significant for many neuroscience modeling tasks.

**Weaknesses:**

Fitting the model jointly to neurons that are not simultaneously recorded (and not necessarily from the same animal) seems to be conceptually at odds with the goal of capturing trial by trial variability. By definition, neurons from different sessions should be considered different for trial to trial variability. I would argue that a more accurate term for the loss is "condition to condition" variability. The authors show it is possible to capture responses across conditions using their objective. It would be great for the authors to expand on this point.

**Questions:**

* Session stitching. Can the authors test their method for session stitching in a simulation, where they stitch together neurons recorded in different sessions as is done in the real data? This would help guide when to trust the results of stitching together non-simultaneously recorded neurons.

* When computing the trial target $T^\prime_{trial}$ for neurons from different sessions, are the trials matched based on condition before summing neurons from different sessions? For example, `Hit` trials are combined from different sessions, but not `Hit` trials and `Miss` trials.

**Limitations:**

Yes

---

> ### Author Rebuttal · Authors · 2023-08-09
>
> We would like to thank the reviewer for reading our paper and providing useful feedback.
> Given the second question of the reviewer, we wonder whether there might be a misunderstanding on the nature of the trial-matching loss function. It appears that the reviewer understands that we only model "condition-to-condition" variability and not "trial-to-trial" variability. But in fact, the concept of trial condition (hit/miss/etc...) is only used for measuring the validity of the model after optimization, and the concept of condition does not appear in the definition of the trial-matching loss function or any other algorithmic step. The trial-matching loss function is only a distance across distributions which happens to capture the trial types among other things. This is the reason why we claim the non-trivial important result that our model can generate the correct distribution of simulated trial types (correct proportion in Figure 4A and correct co-variability of activity in the recorded areas in Figure 4B).
>
> Below we respond to the weaknesses and questions raised by the reviewer:
>
> - *“Fitting the model jointly to neurons that are not simultaneously recorded (and not necessarily from the same animal) seems to be conceptually at odds with the goal of capturing trial-by-trial variability.”*
>
> Following the tradition of rodent research, we consider that we study a system (the sensory-motor pathway) for which the features are shared among animals of the same species and seek to understand the common feature of the recorded circuits and not their differences. For this reason, the implicit statistical assumption of our model is that each session corresponds to a different electrode placement in the same circuit. Since the point by the reviewer is linked to some fundamental modeling choices, we are eager to clarify this in the paper.
>
>
> - *“By definition, neurons from different sessions should be considered different for trial to trial variability.”*
>
> We are not sure if we correctly understand this remark. In the model, neurons from different sessions are indeed considered differently for the trial-to-trial variability. During training for the calculation of the loss function, we consider that neurons from one session are evolving while considering the neurons from the other sessions “hidden”. The loss is constrained on the neuron-specific trial averaged data, every neuron sees the rest as “hidden”, and on the session-specific trial-specific population averages, neurons from one session see the rest of neurons as “hidden”.
>
> - *“I would argue that a more accurate term for the loss is "condition to condition" variability. The authors show it is possible to capture responses across conditions using their objective. It would be great for the authors to expand on this point.”*
>
> As explained above, we believe that it would be reductive to limit the effect of the algorithm to trial types of conditions. We are glad that the reviewer appreciates seeing the responses of neurons across conditions. If you think of specific additional material to clarify our algorithm, please let us know.
>
> - *“Session stitching. Can the authors test their method for session stitching in a simulation, where they stitch together neurons recorded in different sessions as is done in the real data? This would help guide when to trust the results of stitching together non-simultaneously recorded neurons.”*
>
> We agree that a large-scale model recovery experiment would be great to quantify how closely the algorithm is capable of capturing the network structure despite the limited number of recorded neurons per session which is compensated with session stitching. At this point, we can mainly say that session stitching was crucial to leverage all the recorded data available and reconstruct a model of the entire sensory-motor pathway although we never had more than two areas recorded simultaneously.  This would not have been possible without session stitching.

---

> > ### Comment · Reviewer_Cbej · 2023-08-19
> > **Follow up to author response**
> >
> > I thank the authors for their thorough response. It appears I misunderstand how the trial matching loss is computed. As written, I thought trial statistics in eqn. 6 for summed across all neurons for a given area, whether or not these neurons were simultaneously recorded.
> >
> > It makes sense that the recorded statistics in eqn. 6 are computed as sums across the recorded neurons for that trial. For the generated statistics, are those computed as sums across only the neurons that are observed in a session?
> >
> > In your response, you describe neurons as being "hidden" during optimization. Can you elaborate on this point in the paper? I do not see the word "hidden" being used to describe the optimization in the paper (outside of architecture descriptions for the number of hidden units in an MLP).

---

> > > ### Author Response · Authors · 2023-08-20
> > >
> > > Thank you for engaging in the review and the rebuttal of our paper.
> > >
> > > > "It makes sense that the recorded statistics in eqn. 6 are computed as sums across the recorded neurons for that trial. For the generated statistics, are those computed as sums across only the neurons that are observed in a session?"
> > >
> > > Yes exactly. To clarify this we plan to add a session index in the notation T_A^S to indicate that the average over neuron indices recorded in area A during session S. The equation line 161 should be corrected accordingly to make this clear. In this way, we can compute the same statistics in the generated data and in the data recording in session S.
> > >
> > >
> > > > In your response, you describe neurons as being "hidden" during optimization. Can you elaborate on this point in the paper? I do not see the word "hidden" being used to describe the optimization in the paper (outside of architecture descriptions for the number of hidden units in an MLP).
> > >
> > > Sorry for using this term in the rebuttal, we think this term can be slightly ambiguous and avoided it on purpose in the main text. By "hidden" neurons, we mean here the neurons that are not "visible" (aka. recorded) neurons from this session. As we loop over all the sessions, all the neurons are eventually visible, so there is no "always hidden" neuron. We believe this was already clear in the main text (lines 121 - 132), let us know if you think it needs clarification.

---

> > > > ### Comment · Reviewer_Cbej · 2023-08-21
> > > >
> > > > Thanks for the quick clarification. I think writing the full objective across sessions with indices to denote how the terms depend on each session will be very helpful. I will raise my score with this improved understanding.

---

### Official Review · Reviewer_Rsk5 · 2023-07-07

**Soundness:** 3 good
**Presentation:** 3 good
**Contribution:** 2 fair
**Rating:** 5
**Confidence:** 2

**Summary:**

The authors present a method to fit recurrent spiking neural network models to experimental data, extending prior work to multi-session recordings that involve different subpopulations in each session. To capture single trial variability, they introduce a trial matching loss function. The technique is applied to neural recordings from six cortical areas in the sensory-motor pathway during a delayed whisker detection task. The model successfully captures trial-averaged firing rates of individual neurons and population averages. In addition to the expected matching results enforced by the loss function, the model reveals an unexpected mode of variability related to movements during the delay period, resulting in two distinct clusters within hit trials.

**Strengths:**

- Models trial variability beyond trial-averaged firing rates.
- Considers a large-scale, cortex-wide multi-region dataset.

**Weaknesses:**

- Analysis of single trial variability still relies on population averages rather than individual trial averages.
- Lacks a quantitative comparison to other methods.
- Represents an incremental contribution, with the closest related work being Bellec et al., NeurIPS 2021.

**Questions:**

-    Are all results based on held-out data? If so, how was the data held out?
-    What are the new predictions that are not expected results from the training method? Aren't the four main trial types obvious due to the optimization objective including sensory neurons that discriminate between stimulus present/absent and jaw movements that discriminate between lick/no lick, resulting in true positive, false negative, false positive, and true negative categories?
-    If you hold out all neurons from one or multiple entire areas, can your model accurately predict the activity in those areas?
-    Since you use population averages to study trial variability, wouldn't a simple population-level approach using six units, each representing a cortical area, yield the same results?

**Limitations:**

The authors acknowledge that this paper represents only a step towards their long-term goal of developing biophysically interpretable mechanistic models.

---

> ### Author Rebuttal · Authors · 2023-08-09
>
> We would like to thank the reviewer for evaluating our work. We reply to each weakness and question below.
> Weaknesses:
>
> - We are not sure that we understand correctly what the reviewer meant by "individual trial averages". In case the reviewer meant neuron specific individual trials, we tried to fit our model with neuron and time specific features in the form of the trial-matched likelihood as shown in the appendix section B, however it failed to fit the trial-to-trial variability. This most likely happens due to the noise of spiking data.
>
> - Regarding comparison with other methods, in case this was overlooked by the reviewer, we made comparisons between trial matching versus spike-GANs and likelihood-like methods in the discussion and the appendix section B, see Figure A2.
> During the rebuttal period, we have prepared an additional comparison to provide quantitative results between our method and LFADS, the results are provided in the PDF attached to the general rebuttal. If the model assumption can find a model hypothesis that can generate the recorded activity, it seems that our method is not less efficient than LFADS.
>
> - This paper was indeed an inspiring reference for our work, but here we explore and solve new problems (e.g. modeling trial variability and considering a cortex-wide multi-region dataset, the two strengths that are highlighted by the reviewer himself/herself). To be precise, the main methodological innovations of this paper which were not present in [3] are:
>     1) the dataset that was used in [3] was a single recorded session with a single recorded area. The method presented here is scaled to six areas and 28 recording sessions which raised multiple theoretical and practical problems.
>     2) The statistics used in ref [3] were all trial averaged and the method was incapable of producing a realistic distribution of trial-to-trial variability which is central to this paper.
>     3) There are many additional subtleties detailed in our paper/code that were not performed in ref [3]. To cite a few: we use excitatory and inhibitory neurons which are matched coherently to the excitatory/inhibitory cell types detected from the action potential waveform; the weight matrices are restricted to respect Dale’s law; the synaptic delays are restricted in a plausible range and the voltage dynamics follow LIF dynamics with plausible time constants.
>
> Questions:
> - Yes, as stated in the paper lines 214, 216, 226 and in the caption Figure 4B all the reported results are always quantified on the held-out test trials. Those test trials consist of 25% of the trials which are randomly selected from each session. To clarify this, we suggest adding to the paper a sentence to describe the selection of the held-out test trials.
>
> - The split between the four trial types serves in our paper as an explicit way to quantify our success at generating the right distribution of trials. Further, our results go beyond showing "just" the emergence of trial types related to stimulation or movement: 1) the network dynamics are broadly consistent with all trial types and in all areas (see Figure 4B). 2) We have also reported in Figure 4 C-D and between lines 248 and 276 how trial-matching can capture other sources of trial variability like spontaneous motion which is independent of the trial condition structure. Independently of trial types, our method captures all the main modes of trial variability: it was a truly coincidental observation that the highlighted cluster in 4C represented active and quiet hit trials which happen to be trials that were studied in the initial dataset.
>
>
> - Predicting activity in unrecorded areas is an interesting and open question. To some extent, we would like to argue that our model does it in the following sense: given a specific session of the same task where for example only areas wS1 and wS2 are recorded, our model appears to be capable to interpolate faithfully the remaining network activity. This is supported in part by Figure 4 B-C: if the recordings from wS1 and wS2 are sufficient to match the trials relevant to the simulated trials, the corresponding simulated trials will share a correct trial type and many other features reflected in other areas (see Figure 4D).
>
>  - A population-level model would be a fundamentally different model. Given that we take the biophysical interpretability of the model very seriously in this paper, the relevant population level would ideally be a biophysical mean-field like process. Deriving precise and convincing mean-field models is theoretically difficult and -- despite encouraging progress [33] -- we are not aware of a data fitting method using accurate mean-field models with convincing biophysical interpretability.
>
>     However, to demonstrate that a population model would work at the price of many advantages of our model, we tried to sketch out a sensible "simple population level" model. Each area is represented by one or few recurrent tanh units, and these units are mapped to the recorded neurons from the corresponding areas with a linear projection. The reviewer can see the results of our implementation in the General rebuttal. In short, a single tanh unit per area was not enough to yield a competitive generative model even on the artificial dataset. More importantly, this model compromises all the biophysically interpretable aspects of the model (e.g. spiking communication, cell types, Dale's law, and neuron-to-neuron mapping with the data) and we do not know what simple model could easily reconcile that.

---

> > ### Comment · Reviewer_Rsk5 · 2023-08-17
> >
> > Thank you for your clarifications. You understood correctly; I meant neuron-specific individual trials. I appreciate the additional simulations that demonstrate how a simple population-level model not only lacks biophysical interpretability but also fails to capture the data.
> > Whether I increase my score to 6 or not doesn't appear to have a significant impact, considering the reviewers' (unusually) high level of agreement. While this paper might not be the most thrilling, it also doesn't have any significant flaws.

---

### Official Review · Reviewer_Myz7 · 2023-07-07

**Soundness:** 3 good
**Presentation:** 4 excellent
**Contribution:** 3 good
**Rating:** 6
**Confidence:** 4

**Summary:**

This paper presents a framework for training a large recurrent spiking neural network on multi-session recordings by leveraging
an optimal transport-based trial matching between the real data and generated data. This model is use to model the cortical sensory-motor pathway during a tactile detection task.

**Strengths:**

1. In neuroscience, the challenge of training a model on multiple sessions is very relevant. It is expected to record data over multiple sessions and across multiple animals, hence having tools to jointly analyze the underlying dynamics of the neural population across different regions and recordings is important and critical. The use of trial averaging, as motivated by the authors, can be limiting and fails to capture the trial variability during complex behavior. The model presented in this work enables the analysis of the neural population dynamics. The design of the model is clearly well thought-out but remains very simple and is strongly supported by 1) good arguments 2) great visualizations 3) supporting empirical results.
2. The empirical results support the effectiveness of the model at revealing the underlying modes of trial variability. The unsupervised discovery of new modes is also promising.
3. The discussion section is very thorough. In particular the comparaisons with LFADS are interesting and insightful. This work challenges the idea that low-dimensional spaces are required to be able to interpret neural dynamics.

**Weaknesses:**

1. The main weakness of this work is that it was not directly compared to other baselines or test on other recordings during a different behavioral task.
2. The model requires the selection of multiple hyperparameters. One in particular is the number of neurons (1500). The total number of neurons across all recordings is 4415 so there is a huge decrease in the number of units that are modeled. It is unclear how this choice can be made as the size of the dataset increases or for different datasets. The current model takes 3 days of training, are there expected computational limitations for the number of neurons that can be modeled?

**Questions:**

1. While the proposed model is promising, there are questions about how the model can be scaled to larger sets of recordings, as well as more heterogeneous recordings (example: groups of animals differentiated by age, state, disease propagation etc...) Would a single model which ignores the individual differences be adequate?
2. Most neural recordings are currently structured by trials. To truly capture the full breadth of neural code complexity, it is necessary to study these neural dynamics in more unconstrained / complex / free-behavior settings, in which case the notion of trial no longer exists. Might the authors have ideas on how their approach can be adapted to such settings?

**Limitations:**

No limitations identified.

---

> ### Author Rebuttal · Authors · 2023-08-09
>
> We would like to thank the reviewer for the detailed review, and the positive feedback. Regarding the weaknesses of the paper raised by the reviewer:
> 1. About the comparison to other baselines, in case this was overlooked by the reviewer, we would first like to highlight that we had compared trial matching with spike-GANs and likelihood-like approaches in the discussion and the appendix section B, see Figure A2. As far as we know, these algorithms were the only two alternatives capable of guiding the gradient-based optimization of a spiking activity generator like ours. We believe these were the best references to benchmark the efficiency of the trial-matching loss function.
>
>     We agree that the application and comparison of our method to other tasks would be insightful. During the time available for the rebuttal, we prepared an extra simulation and comparison with the LFADS baseline. This new comparison highlights the difference between our biophysically interpretable models and typical deep learning methods. Figure 1 of the attached PDF shows our results. We generated an artificial dataset where area A1 always responds acutely to a stimulus while area A2 responds 80% of the time after a long delay (100, 200 and 300ms). Our methods achieve Pearson correlations similar to LFADS for the 100ms delay showing the efficiency of our numerical method. A striking difference emerges with the larger delays: when the assumption of our network model cannot provide a plausible circuit hypothesis, the trial-matched Pearson correlation saturates at a lower level. In contrast, the performance of LFADS is not affected, which is expected because the computation is not meant to model a plausible network computation.
>
> 2. In this dataset, we chose the number 1500 because, as written in line 119 in the paper, 1) we wanted to have an equal number of neurons per area, and 2) the area with the least number of recorded neurons is wS1 with 258 units (see ref [9]). So, we took 250 neurons per area. Also, the total number of units from the areas of interest is 2724, while the number of 4415 that we had written includes areas that we did not consider to be relevant in our simulation. We agree that this was not clear and should be clarified in the paper.
>
>     We agree with the reviewer that it will be important to scale up our method to larger datasets. So far, we have not had to invest efforts to improve the simulation efficiency, but it seems that the bottleneck to scale for large datasets would be to overcome the memory limit of a GPU. We believe multiple implementation strategies could help. For instance, we currently use full backpropagation through time over 700 timesteps which could be split into shorter time segments using truncated backpropagation. We could also optimize the state representation to be more memory efficient when integrating the network dynamics. Overall, we think that there is a broad margin for improvement.
>
> Questions:
> 1. As explained above, we believe our method can scale to larger networks. Regarding modeling cross-subject differences, it is commonly assumed when using a dataset with dozens of different rodent subjects that the circuit being studied is shared across subjects. We make the same assumption in our model. However, our method also offers possible ways to compare groups of animals differentiated by age (or other features mentioned by the reviewer), by fitting distinct models for subsets of subjects.
>
> 2. We believe that our method could easily be adapted to behavior-free recordings where the trial structure is not straightforward. The notion of trial is only used in our method to define a basis of time alignment to construct the feature vectors. If trials are not defined in the dataset, one could for instance generate "pseudo-trials" aligned to specific events (e.g. onset of jaw movements and/or peaks of activity in sensory areas) or consider other features.

---

> > ### Comment · Reviewer_Myz7 · 2023-08-19
> >
> > Thank you for addressing the points I raised! I particularly like the additional simulation that nicely illustrates the difference between the proposed method and the LFADS baseline. I would like to note that using the default hyper-parameters for LFADS might not be ideal, LFADS can have widely different results for different hyperparameters (AutoLFADS, Keshtkaran et al., 2022).
> >
> > While there are a few limitations including computational efficiency challenges and having to reduce the number of neurons to make sure that all areas have an equal number, I believe that the approach is interesting. I will keep my score at 6.

---

### Official Review · Reviewer_Ec9F · 2023-07-10

**Soundness:** 3 good
**Presentation:** 3 good
**Contribution:** 3 good
**Rating:** 6
**Confidence:** 3

**Summary:**

This paper extends previous works on generative model of neural data using a recurrent spiking neural network.
The main contributions are design of new loss function based on optimal transport to match the trial-to-trial variability in the data, and the approach to extend RSNN generative model to leverage recordings from multiple sessions.
The proposed method is evaluated on both synthetic and real neural data, and authors showed that the method can be used for a hypothesis-free identification of modes of trial-to-trial variability.


**Strengths:**

- Potential usage of the method in answering neuroscience questions is appropriately illustrated from the real data results
- Extension of existing RSNN to data constrained model is novel, and will be valuable step in increasing the usability of related methods in neuroscience.
- Previous work is appropriately discussed and the contributions are clearly stated




**Weaknesses:**

- Although the authors mention in the discussion section the difference in the approach between the proposed method and existing methods, the (potential) advantage of the proposed method (biophysically interpretability) is not clearly illustrated.





**Questions:**

- Definition of $T_{trial}$ in trial matching, Equation 6, and template matching,  line 256, seems to differ.  Is subset of $T_{trial}$ (containing averaged response of area recorded in the particular session/trial) used to compute $L_{trial}$?
- (typo)  $T_{trial}$ → $\mathcal{T}_{trial}$  (line 222 and 244)

**Limitations:**

Yes

---

> ### Author Rebuttal · Authors · 2023-08-09
>
> We would like to thank the reviewer for reading our work in depth. We address below the main weakness and questions raised by the reviewer.
>
> - *"Although the authors mention in the discussion section the difference in the approach between the proposed method and existing methods, the (potential) advantage of the proposed method (biophysically interpretability) is not clearly illustrated."*:
>
> To illustrate more clearly the potential advantage of our method we prepared an extra simulation during the rebuttal period. In Figure 1 shown in the PDF attached to our general rebuttal, we generated an artificial dataset with two putative areas A1 and A2. Area A1 always responds acutely to a stimulus while area A2 responds 80% of the time with a long delay (100, 200 and 300ms). A deep learning method like LFADS [20] can fit and generate the activity perfectly in all cases. For our method, we see that the fitting is as good as LFADS when the delay between the two areas is within a plausible range, and collapses when the network assumption of two areas connected with short 4ms synaptic delays and realistic membrane time constants makes it impossible to produce the activity. This highlights that a successful fit with our model provides a plausible circuit hypothesis relying solely on the recorded neurons.
>
> -  *"Definition of  $T_{\mathrm{trial}} $  in trial matching, Equation 6, and template matching, line 256, seems to differ. Is subset of $T_{\mathrm{trial}}$ (containing averaged response of area recorded in the particular session/trial) used to compute $\mathcal{L}_{\mathrm{trial}}$?"*
>
> Thank you for spotting this subtle difference. Indeed, $T_{trial}$ in equation (6) only considers the areas and neurons which were recorded in a specific session S, while in line 256, as used in the UMAP plot, we compute the feature $T_{A}$ from all neurons from area A of the simulated network. To clarify this in the main text we suggest distinguishing the two by adding a subscript S when $\mathcal{T}_{trial}^S$ contains only the features recorded in session S and explaining the difference.
>
> - Concerning the typo reported by the reviewer this will be corrected.

---

> > ### Comment · Reviewer_Ec9F · 2023-08-17
> >
> > I thank authors for clarification and additional simulation results. Given the response as well as other reviewers comments, I will keep my score at 6.

---

### Author Rebuttal · Authors · 2023-08-09

We would like to thank all reviewers for reading our paper in depth and providing insightful feedback. All reviewers understood the paper and wrote that sections were "clear and novel" and "appropriately discussed and clearly stated". Three of the reviewers commented that the submitted manuscript was lacking baseline comparisons. This could be partly explained because our comparisons with the spike-GAN and the likelihood-based approach were in the discussion and in the appendix, and, in the revision, we will point to them more clearly.  We also ran extra simulations during the rebuttal to compare with other baselines and tackle important questions raised by the reviewers: 1) a new simulation will illustrate the difference between our "biophysically interpretable" model and other approaches, as requested by reviewer Ec9F and 2) we added a population-level model baseline as requested by reviewer Rsk5. The new simulations and associated results are displayed in the PDF attached to the present rebuttal. We are grateful to the reviewers for their comments which help improve the quality of the manuscript.

**New simulation 1: comparison with uninterpretable generative models.** To illustrate the fundamental difference between biophysically interpretable methods against more abstract methods like LFADS, we show in the PDF attached to the rebuttal that our method may only find a functional circuit hypothesis when there exists a biologically plausible circuit hypothesis that can solve the task. We generated a variant of the previous artificial dataset but assumed that the putative motor area (A2) has a bump of activity that arrives with a variable delay from the putative sensory area (A1). When the delay is very large (i.e. 300 ms), a network with direct synaptic connectivity of 2-4ms delay would not be able to bridge this delay, see Figure 1 in attached pdf. So the optimization collapses for large delays and our method does not provide any network solution which explains the artificial data. In contrast, for a method like LFADS, since there is no explicit causal or biophysical interaction that is explicitly required between the two areas, the activity of both areas is well explained independently of the delay and even the input. For short delays, however, we found that our network is as good as LFADS for the metrics that we considered (LFADS was implemented with the tutorial https://github.com/snel-repo/lfads-tutorial, most of the hyper-parameters were the default).

**New simulation 2: comparison with simpler population-level models.** To comment on the difference with population-level modeling, we tried to sketch out a sensible “simple population level” model with crude and imprecise approximations. Compromising many biophysically interpretable aspects of our model (e.g. spiking communication, cell types, Dale's law, and neuron-to-neuron mapping with the data), we have now implemented a spiking activity generator driven by a tanh-RNN with a time constant of 10ms. Each cortical area is represented by one or a few of such units, and the units are recurrently connected to model the interaction between areas. The model generates spikes as Poisson spike trains with a linear projection of the unit's activity to the corresponding area (note that true mean-field models are much more complex [33]).
Testing this model on our artificial dataset, we noticed that one unit per area is not enough to model the two putative areas – it could not explain fully the trial-averaged statistics and failed to capture single trial statistics. It becomes closer to our results using 2 tanh units per area as summarized in Figure 2 in the attached PDF. On the real dataset, we expect that this population-based model might predict activity well given many more than 1 tanh-unit per area. In our mind the main failure of this type of population-level model is not the quantitative fit, rather, there is a tanh-unit that has no explicit mapping with any biophysical substrate. We would argue that this type of model has unfortunately no satisfactory meaning and blurs the line further between biophysically interpretable models and deep learning models like LSTM or transformers. We believe that our choice of neuron- and spike-level modeling is "a sweet spot" enabling simultaneous biophysical interpretability and, as we show, reasonably efficient optimization for parameter fitting.

---

### Decision · Program_Chairs · 2023-09-21

**Decision:**

Accept (poster)

**Comment:**

The reviewers were unanimous in their appraisal of this paper's contributions as above the bar for acceptance to NeurIPS, and I congratulate the authors on their detailed and comprehensive rebuttals, which were critical in leading some reviewers to raise their scores.  I'm pleased to report that this paper has been accepted to NeurIPS.  Congratulations!  Please revise the manuscript to address all reviewer comments and questions.